# Civil Engineering and Malaria Risk: A Descriptive Study in a Rural Area of Cubal, Angola

**DOI:** 10.3390/tropicalmed8020096

**Published:** 2023-02-01

**Authors:** Eva Gil Olivas, Andreu Bruguera, Arlete Nindia E. Eugenio, João José Nunda, Armindo Tchiyanga, Fernando Graça Ekavo, Adriano Cambali, Milagros Moreno, Cristina Bocanegra García, Maria Luísa Aznar, Fernando Salvador, Adrián Sánchez-Montalvá, Israel Molina

**Affiliations:** 1International Health Unit Vall d’Hebron-Drassanes, Infectious Diseases Department, Vall d’Hebron University Hospital, PROSICS Barcelona, Universitat Autònoma de Barcelona, 08035 Barcelona, Spain; 2Hospital Nossa Senhora da Paz, Cubal 690, Angola; 3Emergency Department, Hospital de la Santa Creu I Sant Pau, 08041 Barcelona, Spain; 4Institut d’Investigació Biomèdica Sant Pau (IIB SANT PAU), 08041 Barcelona, Spain; 5Preventive Medicine and Epidemiology Department, Vall d’Hebron University Hospital, 08035 Barcelona, Spain; 6Hospital Municipal, Cubal 615, Angola; 7Public Health Department, Cubal 610, Angola; 8Centro de Investigación Biomédica en Red de Enfermedades Infecciosas (CIBERINFEC), Instituto de Salud Carlos III, 08029 Madrid, Spain

**Keywords:** malaria, Angola, Benguela, incidence rate, civil engineering

## Abstract

(1) Background: Angola is among the high-burden countries with malaria cases globally. After 2013, we suspected an increase in the number of malaria cases in Cubal (Angola), previously in decline. Our objective was to evaluate the incidence rate in Cubal, overall and by neighborhood, for 2014, 2015, and 2016. (2) Methods: A retrospective, observational study was performed in Cubal (Angola) from January 2014 to December 2016, including all patients with a microbiologically confirmed diagnosis, treated at Cubal’s Hospitals for this period of time. The principal variables calculated were the incidence rates of 2014, 2015, and 2016 in Cubal (overall and by neighborhood). (3) Results: There were 3249 malaria cases. The incidence rates were 2.27, 10.73, and 12.40 cases per 1000 inhabitants in 2014, 2015, and 2016, respectively. In the neighborhood, Hamavoko-Kasseke, there was a 10.73-fold increase in incidence during this period. Additionally, Hamavoko-Kasseke presents an anomalous distribution of malaria cases. (4) Conclusions: We observed an increase in the incidence of malaria in Cubal during the three-year study period. The case distribution was highly heterogeneous with hyperendemic areas, and we found a chronobiological association between the construction of a civil engineering project. This information could be useful for deciding which malaria control strategies must be implemented in this area.

## 1. Introduction

Malaria, described by the World Health Organization (WHO) as the deadliest mosquito-borne disease, represents an important global public health challenge. In May 2015, the WHO established the goal of reducing the mortality burden of malaria worldwide by 90% with a deadline of 2030, and although substantial progress has been made in reducing malaria in recent years, it is still present in 106 countries [1,2]. In 2020, due to changes in malaria prevention, diagnosis, and treatment services related to the COVID-19 pandemic, the number of reported cases of malaria increased globally [3]. Between 2020 and 2021, malaria cases continued to rise, although at a slower rate than that observed between 2019 and 2020 [2].

Malaria still represents a significant public health challenge, with an estimated 247 million cases and 619,000 deaths from malaria worldwide in 2021. Moreover, malaria remains endemic in all six WHO regions; most notable is the African region (AFRO), with an estimate of 234 million cases of malaria and 593,000 deaths in 2021, which accounts for 95% of all cases and 96% of all malaria deaths, representing a 32% increase and a 6% decrease compared with 2010, respectively. Three countries in the region accounted for more than 80% of the estimated cases: the Democratic Republic of the Congo (53%), Angola (15%), and Cameroon (12%) [2].

Angola, located in southern Africa, is among the high-burden countries, accounting for 3.4% of all malaria cases globally in 2021, with an estimated 44% increased incidence compared to 2015 [2,3] (Figure 1). Malaria is endemic nationwide, and the risk of malaria affects the entire Angolan population, which totals 32.1 million inhabitants. In fact, malaria is the leading cause of medical care in Angola, along with work and school absenteeism, while being directly responsible for 74% of deaths in children under the age of 5 in 2021 [4]. 

The northern half of the country has a dry season and rainy season; the southern half of the country and the coastal region are semi-arid. These variations in climatic conditions across the country make malaria transmission very heterogeneous throughout the country. Malaria is hyperendemic in the northeast provinces, mesoendemic with stable transmission in the central and coastal ones, and highly seasonal in the four southern provinces [5]. *Plasmodium falciparum* is responsible for more than 90% of malaria infections in Angola, and the anopheline species most involved in transmission are *Anopheles gambiae, Anopheles funestus*, *and Anopheles melas* [5].

Since the end of the civil war in 2002, Angola has made great strides towards achieving malaria control, including the establishment of the current national guidelines, indoor residual spraying of selected urban districts, free distribution of insecticide-treated nets (ITNs), free Artemisinin-based combination therapies (ACTs) at public health facilities, and preventive malaria treatment for pregnant women [4,6]. However, in 2015, coinciding with Angola’s financial crisis and the cessation of support for malaria control from the Global Fund, there was a drastic drop in ITN procurement and a country-wide stock-out of ACTs and rapid diagnostic tests (RDTs) [7,8]. In addition, in 2020, Angola suffered a health crisis experienced worldwide because of the COVID-19 pandemic, which slowed down surveillance measures for other diseases, including malaria.

Therefore, the National Malaria Control Program of Angola (NMCP) estimated that there were 9.2 million malaria cases in the country during 2021, representing a 20.4% increase between 2017 and 2021 [4]. 

Cubal, with a population of just over 320,000 inhabitants (according to the only Angolan census), is a rural municipality located in the province of Benguela (Figure 1). Most of the population does not have access to drinking water, electricity, or sanitation, with agriculture and livestock being the main activities [9,10].

Malaria transmission in Cubal is reported to be historically mesoendemic stable, with a six-month rainy season from September to April; however, there are no published data on the transmission in this area [9,10]. 

Between 2014 and 2016, the reported incidence rate was >1700 cases per 100,000 inhabitants; by year (2014, 2015, and 2016), the case rates were 1778, 1835, and 2719, respectively, according to data provided by the Department of Public Health of Cubal.

From 2009 to 2013, there was a downward trend in the number of malaria cases in Cubal. Possible reasons for the decrease in malaria in this area of Angola may include socio-economic changes, the implementation of prevention measures against malaria, and possible changes in environmental factors, such as rainfall [11]. 

Although what would be expected would be a continuous decrease in subsequent years, the perception of health professionals in this municipality after 2013 is that there was a progressive increase in the number of cases compared to previous years, and this increase has not been the same in all the neighborhoods of the municipality [11]. In order to provide more accurate data, we evaluated the incidence rates in Cubal, overall and by neighborhood, for the years 2014, 2015, and 2016.

## 2. Materials and Methods

### 2.1. Study Design

This was a retrospective, observational study designed to assess the incidence of malaria in Cubal, Angola (overall and by neighborhoods) from January 2014 to December 2016.

Cubal is located in the highlands of the central plateau of Angola. It is located 146 km from the city of Benguela, the capital of the Benguela province. It has an extension of 4.794 km^2^ [9,10] (Figure 1).

Cubal has two referral hospitals, the Hospital Municipal (HM), a 177-bed public hospital located in the center of Cubal Sede, and the Hospital Nossa Senhora da Paz (HNSP), a privately managed 300-bed hospital integrated into the public health system, located on the outskirts of Cubal Sede [9,10].

### 2.2. Study Population and Data Collection

This study included all patients with a microbiologically confirmed diagnosis of malaria treated at the HM and the HNSP from January 2014 to December 2016.

Malaria was confirmed by (1) Giemsa-stained (10%) thick blood film observed under a light microscope by an expert microscopist and/or (2) the Paracheck-Pf^®^ rapid diagnostic test (Orchid Biomedical Systems, India).

Data were collected by reviewing the participating hospitals’ patient registries. We included all patients treated at the hospital (outpatients, inpatients, and those treated in the emergency room). The data were confirmed and completed by checking the record books of the hospital laboratory. Subsequently, these data were compared with data from the Cubal malaria case report for 2014, 2015, and 2016 provided by the Cubal Public Health Department.

Microbiological diagnosis of malaria was only performed in HM and HNSP. For this reason, primary health centers or informal health points were excluded from the analysis. Demographic and clinical data were collected (including age; sex; origin; date of diagnosis; the hospital where the diagnosis was made; diagnostic method (RDT and/or thick blood film); *Plasmodium species*; parasitemia in patients diagnosed by thick blood film; malaria severity criteria; treatment received; follow-up regimen (outpatient or inpatient); and final outcome (discharge, death, or lost to follow-up)). Complicated malaria was defined according to the WHO context-adapted severity criteria [12]. Patients who met at least one of the WHO criteria were considered complicated. In most cases, classification was mainly based on clinical criteria, given that biochemical reagents for creatinine, bilirubin, and hemoglobin were not always available.

Rainfall, temperature, and humidity information for Cubal for the years 2014–2016 were obtained from worldweatheronline.com [13].

Only one country-wide census has been performed to date in Angola in the year 2014. The provisional results of that census were published in March 2016. The census report describes the number of inhabitants by municipality but does not break down the number of inhabitants by neighborhood [9]. Therefore, to calculate the incidence rate by neighborhood, we estimated the number of inhabitants per neighborhood (denominator). This procedure was described in detail elsewhere. Briefly, however, a satellite image obtained from Google maps (v. 9.68) was used. Next, square cells per square kilometer of the image of the city of Cubal were analyzed (Figure 2). Within the selected grid cells, structures of appropriate size and shape were identified as potential households, called family units, and manually enumerated and multiplied by the average number of inhabitants per family aggregate, according to national records [9]. Thus, we constructed a household census for all of the neighborhoods in Cubal. This procedure is similar to that used in other studies with similar demographic characteristics [14,15].

### 2.3. Statistical Analysis

All study data were collected in one database (Microsoft Excel). IBM-SPSS Statistics 25.0 statistical package was used to perform all statistical analyses. Qualitative variables were presented as absolute numbers, and percentages and quantitative variables as means with standard deviation (SD) or medians and ranges, depending on the distribution. The Chi-square test or Fischer’s exact test was used when the expected values were less than 5 to compare categorical variables. Student’s t-test was used to compare quantitative variables (means or medians). If the conditions for parametric tests were not met, non-parametric tests were used. Differences were considered statistically significant at *p* ≤ 0.05.

The cumulative monthly malaria incidence was calculated and stratified by age and geographic area. Next, the neighborhoods were unified by the number of family units and demographic characteristics to obtain a more homogeneous sample. Additionally, incidence rates were compared.

### 2.4. Ethical Considerations

The study protocol was approved by the institutional review board of both hospitals in Angola (HM and HNSP) and by the Ethical Review Board at the Vall d’Hebron University Hospital (PR(AG)383/2016). This study was approved by the governmental organizations of Cubal with the support of the Public Health department, as well as by the “sobas” (traditional leaders in Cubal).

All data were anonymized. Data confidentiality was ensured throughout the study in accordance with the ethical standards of the Declaration of Helsinki.

All the authors declare that they have no competing interests and report no financial support.

## 3. Results

In our study, there were a total of 3249 malaria cases in Cubal between January 2014 and December 2016. Most cases (n = 1649, 51.6%) were observed in patients under age 5; of these, 38.4% (n = 633) were from the Hamavoko-Kasseke neighborhood. Females accounted for 50.7% of cases (n = 1645). Hospitalization was required in 64.4% of cases (n = 1954). Most patients (89.7%, n = 1414) received intravenous antimalarial treatment, and 8.7% died (n = 153).

The diagnosis was mostly made by RDT 50.6% (n = 1551), and only 22.2% (n = 677) of malaria cases were diagnosed by RDT and thick blood film. *Plasmodium falciparum* accounted for nearly all of the diagnosed species (99.7%, n = 2451). Other species included *Plasmodium vivax* (n = 2) in the Hamavoko-Kasseke and Camunda neighborhoods, *Plasmodium ovale* (n = 1) in Hamavoko-Kasseke, and *Plasmodium malariae* (n = 4) in Camunda, Bairro 80, Cristo Rei, and Passagem. Only one infection with *Plasmodium vivax* was solely due to this pathogen; the rest of the infections with *Plasmodium vivax* were coinfections with *Plasmodium falciparum.* The results are shown in Table 1.

Of the 3249 cases, 791 (24.3%) met at least some of the criteria for complicated malaria. Anemia, defined as hemoglobin levels <7mg/dL in adults and <5 mg/dL in children, was the most common criterion (n = 707, 89.4%). Other criteria were cerebral malaria (n = 202, 25.6%), jaundice (n = 24, 3%), renal failure (n = 10, 1.3%), hypoglycemia (n = 9, 1.1%), and respiratory distress (n = 7, 0.9%). Complicated malaria was more common in children < 5 years of age (n = 530, 67%; *p* < 0.001). In several neighborhoods (Hamavoko-Kasseke, Bairro Novo-Marco, Calomanga, As 200-Bairro 80, Kassiva-Tchimbassi, and Cerámica), >30% of cases met the severity criteria. Complicated malaria cases were 5.7% (n =45), 73.1% (n = 578), and 21.2% (n = 168) in 2014, 2015, and 2016, respectively (Figure 3).

The incidence rates by year (2014, 2015, and 2016) were 2.27, 10.73, and 12.40 cases, respectively, per 1000 inhabitants, which is a more than a five-fold increase (5.45) between 2014 and 2016. In some neighborhoods, there was a 7-fold increase in the incidence rate during this period, most notably in Hamavoko-Kasseke, with a 10.73-fold increase (*p* < 0.01) (Table 2 and Figure 2). Given the increased incidence of malaria cases in Cubal, we evaluated changes in rainfall, temperature, and humidity in the city from January 2014 to December 2016. The temperature and humidity data were analyzed without finding significant variations (data not shown). Rainfall increased from September to April (wet season), which was followed by a peak in malaria cases in both 2014 and 2015 (Figure 4).

Figure 5 shows the incidence rate of malaria in the Hamavoko-Kasseke neighborhood, by months, between January 2014 and December 2016. As the figure shows, the distribution was anomalous. We see that the incidence rate in Hamavoko-Kasseke in the month of July 2015 (dry season) was 1.29 versus an average of 0.34 in the other neighborhoods. The low rate observed in most of Cubal is to be expected in the dry season when the incidence rate is low. However, in 2015, the incidence curve in Hamavoko-Kasseke increased from July to November, when it reached an incidence of 2.21. By contrast, in the other neighborhoods, the incidence only began to rise in November, coinciding with the beginning of the rainy season. Although this pattern in Hamavoko-Kasseke was not observed in 2014, it did occur again in the dry season of 2016.

## 4. Discussion

The findings of this study show that the incidence rate of malaria in Cubal (Angola) increased significantly from 2014 to 2015/2016. These changes were highly variable among the neighborhoods in Cubal, suggesting the involvement of neighborhood-specific factors.

According to the 2017 World Malaria Report, the overall incidence of malaria decreased by 20% in the African region from 2010 to 2016; however, between 2014 and 2016, the incidence increased in some countries [8]. This change in trend may be attributable to several factors, including the decline in both international and national funding, conflicts in endemic areas, abnormal weather patterns, and the emergence of insecticide-resistant mosquitos and parasites resistant to antimalarial drugs. This increased incidence rate was observed in numerous countries, including Angola [8,16]. Previously published data from the municipality of Cubal confirmed precisely the increase observed in the number of malaria cases [10].

Salvador et al. observed that Cubal may have a possible seasonal mesoendemic distribution, even though transmission in this region was believed to follow a stable mesoendemic pattern [4,11]. Our findings are consistent with those described by Salvador and colleagues: we also observed a seasonal mesoendemic distribution during the three-year study period, with most cases (52%) occurring in children < 5 years of age. The seasonal transmission of malaria has several important implications, mainly the need to implement control strategies, including chemoprevention in children [16,17]. Given the findings of our study, together with the recommendations made by Salvador et al., we believe that further studies are needed to determine whether Cubal should be considered a seasonal transmission zone.

Our analysis reveals significant spatial variation in malaria transmission, with hyperendemic areas within a mesoendemic city. For example, certain neighborhoods (e.g., Hamavoko-Kasseke) have substantially higher incidence rates (Table 2) than other neighborhoods (e.g., Passagem). The heterogeneity in Cubal is similar to that observed in Dande, in northern Angola, where hyperendemic areas have also been found within a mesoendemic zone [18]. As Tamara et al. observed, the city of Kampala presents a similar pattern [19]. The authors evaluated the factors influencing these variations, concluding that multiple different factors influence the distribution of malaria, which may even present a heterogeneous distribution within the same neighborhood. This finding further underscores the importance of creating malaria distribution maps of small geographic areas in order to design targeted interventions to prevent malaria while also optimizing resources [20].

In our study, most cases were concentrated in the Hamavoko-Kasseke neighborhood. Compared to the other neighborhoods, this neighborhood was unique, showing an atypical temporal/seasonal distribution. In Figure 5, we can observe an increase in the incidence ratio of malaria cases in Hamavoko-Kasseke in July 2015 (the start of the dry season), which is maintained until November 2015, where the increase, characteristic of the start of the wet season, is maintained. The same findings were present in 2016 but absent in 2014. The notable increase in the incidence rate in the Hamavoko-Kasseke neighborhood in 2015, together with the emergence of a seasonal pattern that differed from the other neighborhoods, points to the likely involvement of neighborhood-specific factors.

In Hamavoko-Kasseke, there is a pond on the northern edge of the neighborhood. This is relevant given that proximity to water sources is a known risk factor for the proliferation of anopheline mosquito breeding sites, which is why the risk of contracting malaria increases in such areas [19,20]. Nevertheless, this pond was already present in this neighborhood in 2014 and, thus, unlikely to explain the increased incidence rate. Rather, the unexpected increase in the incidence rate in this neighborhood could be explained by the construction of a new road, which began in February 2015 and was completed in November 2016. The road runs through the entire neighborhood and, crucially, contains a ditch on both sides (about 2 m deep) built to channel water during the rainy season. These ditches lead to water collectors that concentrate and drain the water, potentially promoting the emergence of mosquito breeding sites near highly populated areas (Figure 6). In fact, this finding is in line with other studies in other African countries. For example, in a study conducted in Zambia, Pinchoff et al. discussed how urbanization, the creation of irrigation systems, and road construction can have a major impact on malaria incidence by creating new permanent mosquito breeding sites, which allow mosquitos to breed throughout the year [15,21].

In our study, 50.6% of malaria diagnoses were made by rapid tests and 22% by a combination of thick film analysis and RDT. RDT, although not the gold standard for malaria diagnosis, has 98% sensitivity and 99.3% specificity for P. falciparum against microscopy and is approved for use as a diagnostic test by the WHO [22,23]. Our finding is consistent with those presented by the NMCP and the World Malaria Report 2016 [4,24]. However, these data contrast with some studies conducted in other rural areas of Angola, in which diagnosis was made by RDT alone. One such example is a study by Mateusz et al. conducted in Huambo and Uige in 2016, in which nearly 80% of cases were diagnosed with RDT [25]. The poor sanitary conditions in many parts of rural Africa are well known, which could explain the high percentage of rapid tests used to diagnose malaria in those areas. The NMCP reported that 47% of malaria diagnoses were made by RDT in 2013 versus 82% in 2016 (80% in Benguela) [24].

Notably, most of the malaria patients in our study (89%) received intravenous antimalarial treatment. In addition, 64% required hospitalization, even though only 25% presented criteria for complicated malaria. However, with regard to the last finding, it is important to note that in the years 2015 and 2016, the availability of the reactants needed to perform biochemical and blood count analyses was limited and intermittent. As a result, the definition of “complicated malaria” was, in many cases, exclusively based on clinical judgment, a clear study limitation in terms of diagnostic certainty. This result also shows how both national and international guidelines for the management of malaria are often disregarded, a finding that was also made by Mateusz et al. [25]. Given the lack of continuing education, together with the continual shortage of diagnostic material and limited treatment options, healthcare personnel are forced to manage patients with malaria as best they can in these real-life conditions.

Our findings regarding the percentage of complicated cases differ from those reported by the NMCP, which found that 9.4% of malaria cases in 2015 met the criteria for complicated malaria, and 77% received an oral ACT combination therapy [24]. This indicates a discrepancy between the diagnostic tests performed, the reported cases, and the treatment provided, as reported in other regions of the country [25,26]. These discrepancies, as presented in a paper by Julie I. et al. in Luanda, cast doubt on the true burden of malaria and highlight the need to strengthen the NMCP [27]. As referred to in the WHO’s Test, Treat, and Track (T3), case management and reporting are essential to optimize resources and reduce malaria morbidity and mortality of malaria to address the serious consequences derived from the withdrawal of funds [28].

This study has several limitations, most of which are related to the retrospective design. First, the results were based on data obtained by reviewing hospital registration records and medical records. In any case, we are confident that the data obtained represent a situation very close to the real one because, on the one hand, they are the only existing registries in the city, and on the other hand, the national malaria control program has implemented a data collection system that obliges all centers to send periodic reports each month, thus, reducing the error rate. In any case, we are aware of this limitation that is inherent to the design of the project. Likewise, as mentioned above, reagents for biochemistry and blood count analyses were often out of stock, and thus, the diagnosis of “complicated malaria” was based on clinical judgment.

The epidemiology of malaria is highly complex due to the multitude of factors involved (malaria parasites, vectors, human hosts, and the environment). Notwithstanding these challenges, understanding the link between malaria transmission and climate- and human-related factors is essential to developing measures to reduce transmission and eliminate malaria from endemic areas. Currently, factor mapping of small areas considered hotspots is a widely used technique to optimize resources [18,20,21,26].

## 5. Conclusions

In the present study, we observed a substantial increase in the incidence of malaria in Cubal during the three-year study period. The case distribution was highly heterogeneous across hyperendemic areas, even within the same municipality. Our findings show that Cubal has a seasonal mesoendemic malarial transmission pattern in an area considered historically stable mesoendemic. The seasonal transmission of malaria has several important implications, mainly the need to implement control strategies, including chemoprevention in children [17,29]. Furthermore, an interesting finding of this study was the chronobiological association between the construction of a civil engineering project (a road) and the incidence of malaria.

The construction of a road (and the ditches designed to divert water from the road) was the only factor with chronological and biological plausibility that could have provided favorable conditions for mosquito breeding sites near inhabited areas, leading to a different seasonal pattern compared to other neighborhoods in the district.

Further studies are essential to evaluate the possible factors influencing the observed change in the epidemiology of Cubal, such as factors related to people’s behavior and knowledge, as well as complementary entomological studies.

## Figures and Tables

**Figure 1 tropicalmed-08-00096-f001:**
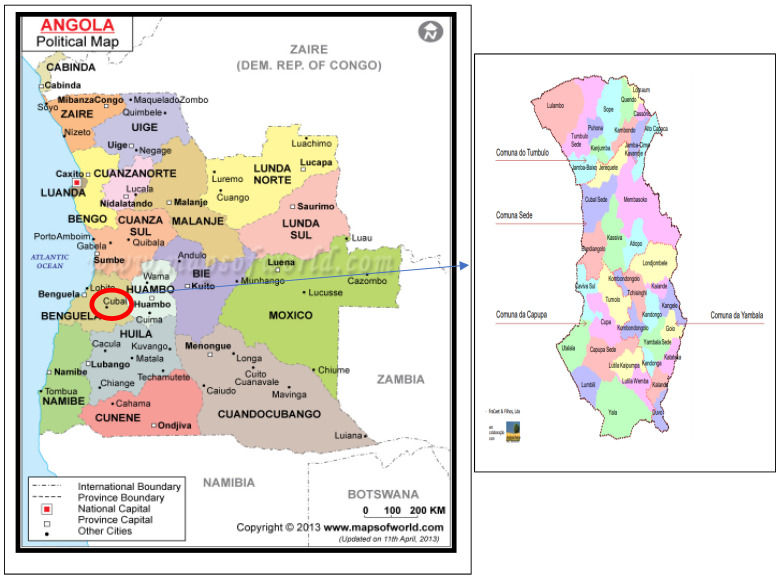
Map of Angola and map of Cubal.

**Figure 2 tropicalmed-08-00096-f002:**
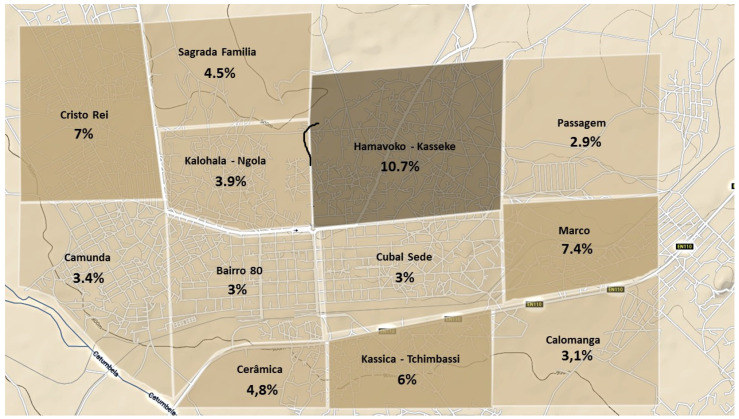
Map of Cubal with its neighborhoods. The fold change in the increase in incidence rate is represented. Cases per 1000 population between 2014 and 2016.

**Figure 3 tropicalmed-08-00096-f003:**
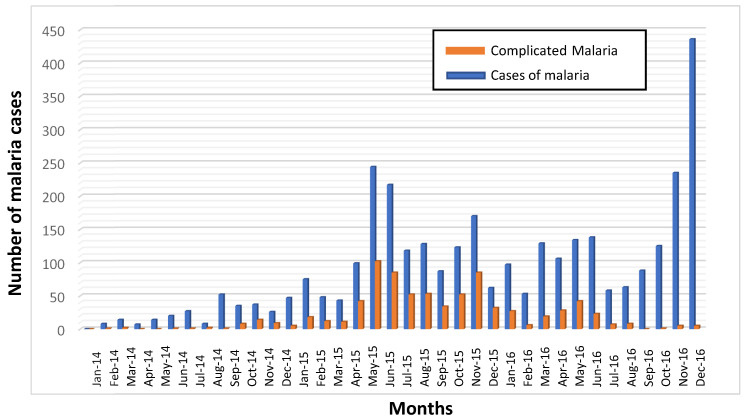
Absolute number of monthly cases of malaria (blue columns) and complicated malaria (orange columns) in Cubal, 2014–2016.

**Figure 4 tropicalmed-08-00096-f004:**
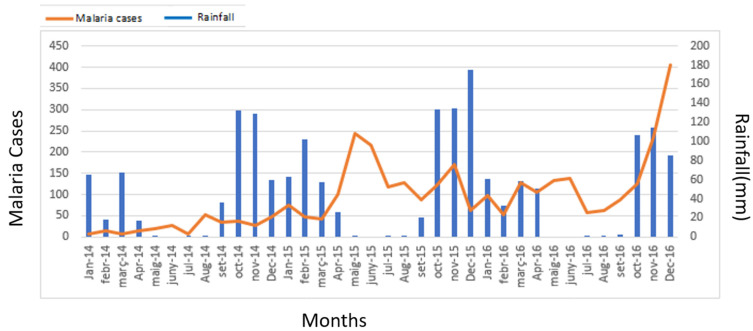
Absolute number of malaria cases (orange) and rainfall (blue columns) by month in Cubal, years 2014 and 2016.

**Figure 5 tropicalmed-08-00096-f005:**
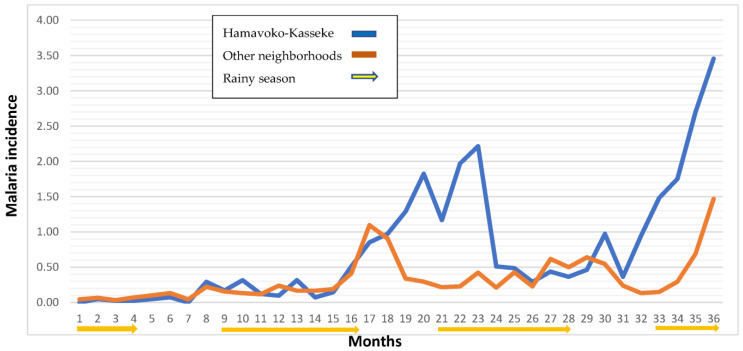
Incidence rate of malaria (cases per 1000 population) in the Hamavoko-Kasseke neighborhood compared to other neighborhoods (January 2014–December 2016).

**Figure 6 tropicalmed-08-00096-f006:**
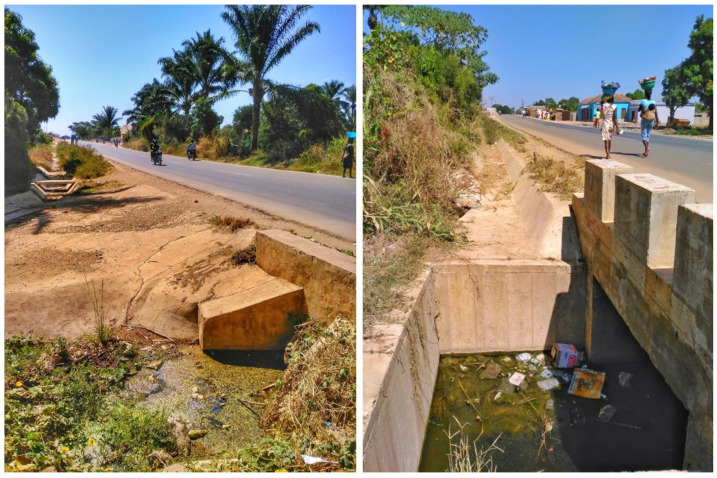
The new road with its drainage water collectors. Observe that in both figures, water is present despite being in the drought season.

**Table 1 tropicalmed-08-00096-t001:** Demographic characteristics of patients diagnosed with malaria in Cubal, Angola, from 2014 to 2016.

	Malaria Cases, N = 3249	Uncomplicated Malaria, N = 2458	Complicated Malaria, N = 791	*p*-Value
**Sex**				0.475
Male	1598/3243(49.3%)	1199/2452 (48.9%)	399(50.4%)	
Female	1645/3243(50.7%)	1253/2452(51.1%)	392(49.6%)	
**Age, years**				<0.001
≤5	1649/3195(51.6%)	1119/2419(46.2%)	530/776(68.3%)	
6–15	1052/3195(33%)	834/2419(34.5%)	218/776(28.1%)	
16–45	434/3195(13.6%)	408/2419(16.9%)	26/776(3.4%)	
>45	60/3195(1.8%)	58/2419(2.4%)	2/776(0.2%)	
**Hospital**				<0.001
HNSP	2661(81.9%)	2061(84%)	600(75.8%)	
HM	588(18.1%)	397(16%)	191(24.2%)	
**Diagnostic method**				<0.001
Thick blood film	836/3064(27.2%)	773/2308(33.4%)	63/756(8.3%)	
Quick test	1551/3064(50.6%)	1209/2308(52.4%)	342/756(45.2%)	
Thick blood film plus RDT	677/3064(22.2%)	326/2308(14.2%)	351/756(46.5%)	
**Parasitemia (thick blood film) ***				<0.001
<10,000	503/760(66.2%)	240/392(61.2%)	263/368(71.5%)	
10,000–100,000	176/760(23.2%)	117/392(29.8%)	59/368(16%)	
>100,000	81/760(10.6%)	35/392(9%)	46/368(12.5%)	
**Species**				<0.001
*P. falciparum*	2451/2458(99.71%)	1827/1830(99.85%)	624/628(99.38%)	
*P. ovale*	1/2458(0.04%) *	1/1830(0.05%) *	0	
*P. vivax*	2/2458(0.08%)	1/1830(0.05%)	1/628(0.15%) *	
*P. malariae*	4/2458(0.18%) *	1/1830(0.05%) *	3/628(0.47%) *	
**Treatment**				<0.001
Artemether/lumefantrine v.o.	163/1577(10.3%)	157/801(19.6%)	6/776(0.8%)	
Artemether iv	1119/1577(70.9%)	585/801(73%)	534/776(68.8%)	
Quinine iv	295/1577(18.8%)	59/801(7.4%)	236/776(30.4%)	
**Regimen**				<0.001
Inpatient	1954/3033(64.4%)	1172/2252(52%)	782(100%)	
Outpatient	1079/3033(35.6%)	1079/2252(4844%)	0	
**Final outcome**				<0.001
Discharged	1614/1767(91.3%)	1101/1148(95.9%)	513/619(82.9%)	
Death	153/1767(8.7%)	47/1148(4.1%)	106/619(17.1%)	

Abbreviations: HNSP—Hospital Nossa Senhora da Paz; HM—Hospital Municipal; RDT—rapid diagnostic test; i.v.—intravenous; v.o.—via oral; * coinfection with *P. falciparum.*

**Table 2 tropicalmed-08-00096-t002:** Incidence rates (cases per 1000 inhabitants) by neighborhood in Cubal, Angola, 2014 to 2016.

Neighborhood	Total	2014	2015	2016	Fold Change From 2014 To 2016 *
Hamavoko and Kasseke	50.97	2.49	21.80	26.69	10.73
Bairro 15, Bairro Novo, and Marco	22.46	1.56	9.28	11.62	7.44
Calomanga	5.63	0.63	2.98	2.01	3.18
Camunda	23.22	3.06	9.63	10.53	3.44
Cristo Rei and Cristo Rei E Benfica	48.20	4.02	15.81	28.37	7.06
Estege, Vila de Cubal, AS 200, and Bairro 80	21.60	3.92	5.93	11.75	3.00
Kalohala and Ngola	22.24	2.97	7.64	11.63	3.91
Kassiva and Tchimbassi	21.97	1.32	12.61	8.03	6.08
Passagem	18.01	2.96	6.44	8.61	2.91
Sagrada and Assunção	75.57	8.98	25.81	40.78	4.54
Tinguita or Ceramica	6.89	0.68	2.97	3.24	4.80
**Total**	24.70	2.27	10.73	12.40	5.45

Hamavoko-Kasseke incidence rate was compared with each neighborhood, and in all of them, the *p*-value was <0.01. * Increase in cases from 2014–2016, calculated by dividing the number of cases in 2016 by the cases in 2014. (Number of times incidence increased between 2014 and 2016).

## Data Availability

The datasets used and analyzed during the current study are available from the corresponding author on reasonable request: egilo@santpau.cat.

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
