# Peer review of "Civil Engineering and Malaria Risk: A Descriptive Study in a Rural Area of Cubal, Angola"

_tropicalmed, 2023, doi:10.3390/tropicalmed8020096_

Round 1

Reviewer 1 Report

This is a retrospective epidemiological study of malaria in Angola. The main interesting and unique point of this manuscript is the chronobiologic association between the civil construction and the incidence of malaria in that area. And the result of this study might add some knowledge on epidemiology and might help policy maker how to control malaria in the area similar to the study sites.

However, I have several questions and concerns.

1. Since this is a single site study in specific area of Angola, it is difficult for readers to understand the local situation. It will help if authors could provide a map of Angola and another map of Cubal area. And where are the two hospitals located in the Cubal area.

2. Table 2 is also difficult to understand, since readers will have no idea where those neighborhoods located. And the environment/geography in those areas are the same or not; some may be the forested area, some may be more urban?

3. The incidence of malaria in this study were calculated by the number of cases from two referral hospitals (HM and HNSP) as mentioned in the study design; is it possible that there are other local hospitals/clinics that treated malaria patients as well. Or all malaria cases needed to be treated in those two referral hospitals. Could you please clarify this?  It will determine the likelihood of underestimation of incidence of malaria.

4. The data used in this study is quite old (2014-2016), do author have more recent data at least before COVID pandemic i.e. 2017,2018,2019.

5. Figure 5 is very interesting. Was there any entomological survey of sampling of that water? It would be great and convincing if authors could demonstrate Anopheles larvae on those water as well.

Reviewer 2 Report

The manuscript was easy to read and understand. Very few typo errors and minor corrections are needed.

As I understand from the manuscript, most of the data used were secondary data due to the study's retrospective study approach. So, there is a limitation on the availability of accurate data to use. But I do think there is a need to add more evidence (examples) and data analysis to convince readers that the reason for an increase of malaria cases in Cubal was due to only one road construction work in 2015 and 2016, as stated in the discussion section. I suggest a more detail analysis of this evidence is needed to make the discussion and conclusion more convincing and compelling.

I think if the authors are able to add more information and data analysis on this particular issue, then it will justify the use of 'Civil engineering' in the title of the manuscript.

List of corrections:

1. Figure 1 - what does FECHA means in the graph?

2. Figure 5 - a scale bar is needed for the map of Cubal

3. Line 331-332 - please delete the last sentence and do please double check the references' formatting as they are not consistent throughout the list.

4. References Line 381- What does number 6 means? 

Reviewer 3 Report

Dear Authors

Although the study of malaria and the role of environmental factors on the occurrence of the disease is very useful in planning for the management of the disease, it is necessary to follow all the principles of writing a scientific article. In this regard, I have made comments in the attached file that I hope will be of your attention.

Good luck

Reviewer 4 Report

The MS is the descriptive study of malaria which has increased from 2014-16. The study shows the incidence of malaria in Cubal by hospital records. Additionally, the author also described the demographic and clinical data of malaria patients. Although the MS is of potential interest for research in the field of malaira, I suggest this MS for publication after minor addition as the following lists:

1. For title, the semi-colon ( : ) need to replace prior to "A descriptive..."

2. The rationale for the changes that made malaria comes back could be added.

3. To my opinion, the author may give the reader an idea about the neighborhood names in terms of the geographic, demographic, etc. 

4. It would be better if (possible) the author can describe the distribution of malaria cases in terms of the spatial distribution and the road construction ( this is the suggestion)

5. In figure 1, the months should be written in english (in x axis).

6. In figure 1, the bar needs to revise the pattern to make it easy to classify for readers. Same as the line in figure 3

7. it is better to mention the neighborhood name on the map in figure 5 and show the significant increasing of malaria cases on the map.

8. RDT sensitivity and specsitivity need to be discussed. 

9. Did the author observe/ survey the breeding site of mosquito anopheline in the channel along the new road? Otherwise, the author can not overclaim the possible risk of malaria vector from this construction and show the photo like in figure 5. 

10. The MS needs to recheck the grammatical.

11. The references must be revised. 

Round 2

Reviewer 1 Report

.

Author Response

ANSWER TO REVIEW 1

POINT 1.

*The main limitation in this paper is the time frame studied. Data is too old
i.e. 2014-2016. So the usefulness of this paper is very limited. It would be
great if author could show the updated one.*

Response 1: We are aware that the data in the article are old.

Several personal reasons have prevented me from analyzing them and writing the manuscript until now.

We believe that the relevance of the article is not the date of the data, just the possible impact that a civil construction can have to modify the epidemiology of malaria in a population. And we believe that this relevance is independent of the date of the data because it has a great impact on the population and on the preventive measures to be adopted.

POINT 2.

In addition, according to reviewer 3's comments, could you please upload two
versions (clean and tracked change)?

Response 2. A version of the manuscript with tracked change  and another version without change control, the clean manuscript, are sent.

Reviewer 3 Report

Dear Authors

Thanks for considering the comments. Really the revised version is in track change format and it is not clear how the changes were implemented. Please send the clean version of the manuscript as well.

Good luck

Author Response

ANSWER TO REVIEW 3

POINT 1.

In addition, according to reviewer 3's comments, could you please upload two
versions (clean and tracked change)?

Response 1. A version of the manuscript with tracked change  and another version without change control, the clean manuscript, are sent.

Round 3

Reviewer 1 Report

 I have no further comment. 

Reviewer 3 Report

Thanks for considering the comments.